# Robust SCKF Filtering Method for MINS/GPS In-Motion Alignment

**DOI:** 10.3390/s21082597

**Published:** 2021-04-07

**Authors:** Huanrui Zhang, Xiaoyue Zhang

**Affiliations:** School of Instrumentation and Optoelectronic Engineering, Beihang University, Beijing 100191, China; zhanghr@buaa.edu.cn

**Keywords:** robust filtering algorithm, MIMU, in-motion alignment

## Abstract

This paper presents a novel multiple strong tracking adaptive square-root cubature Kalman filter (MSTASCKF) based on the frame of the Sage–Husa filter, employing the multi-fading factor which could automatically adjust the Q value according to the rapidly changing noise in the flight process. This filter can estimate the system noise in real-time during the filtering process and adjust the system noise variance matrix Q so that the filtering accuracy is not significantly reduced with the noise. At the same time, the residual error in the filtering process is used as a measure of the filtering effect, and a multiple fading factor is introduced to adjust the posterior error variance matrix in the filtering process, so that the residual error is always orthogonal and the stability of the filtering is maintained. Finally, a vibration test is designed which simulates the random noise of the short-range guided weapon in flight through the shaking table and adds the noise to the present simulation trajectory for semi-physical simulation. The simulation results show that the proposed filter can significantly reduce the attitude estimation error caused by random vibration.

## 1. Introduction

With the development of Micro-Electro-Mechanical System (MEMS) technology, the MEMS inertial measurement unit (MIMU) is gaining attention due to its integrability and miniaturization in several fields such as guided weaponry, unmanned automatic vehicles (UAVs), and robots [1,2,3,4,5,6]. As a dead-reckoning method, the initial alignment accuracy has a vital impact to the final inertial navigation results, while there are only tens of seconds for a tactical weapon. Thus, achieving rapid, accurate, and robust initial alignment is essential to the application of the MIMU. As for the MIMU, it cannot detect the Earth’s rotation properly with high gyro noise, so alignment methods for the high-accuracy Inertial Navigation System (INS) are not suitable for it. A widely used method to achieve initial alignment for the MIMU is to use an extra sensor to provide additional information as a reference for initial alignment during the flight process. The in-flight alignment has been investigated as one of the key technologies for guided weapons and UAV applications in MIMUs in the past 10 years.

The numerous methods to solve in-flight alignment for the MIMU could be mainly divided into two categories, namely the optimization-based approach and the Kalman filter approach. The first one usually turns the problem of alignment into a Whaba’s problem. To solve the problem, several algorithms have been developed, such as the q-method, Quaternion ESTimator (QUEST), and Recursive QUEST (REQUEST) [7,8]. These methods calculate the optimal quaternion by constructing the optimal observation vectors through different filtering methods [9,10,11,12,13]. The latter method is to obtain the misalignment angle between the navigation frame to the computing frame of the INS by constructing an appropriate model for filtering. Owing to the nonlinearity of the misalignment model and the large noise caused by environment, the nonlinear filtering methods occupy a great proportion of the in-motion alignment methods to achieve stability, accuracy, and robustness.

In order to handle with the strong nonlinear system, several filtering methods based on the Kalman filter have been proposed, such as the extended Kalman filter (EKF) [14,15], unscented Kalman filter (UKF) [16,17,18], Gauss–Hermite quadrature filter (GHQF) [19,20], sparse grid quadrature filter (SGQF) [21,22,23], and cubature Kalman filter (CKF) [24,25]. The EKF approximates the nonlinear model to a linear model through Taylor expansion to deal with nonlinear problems. The advantage of this is that the filter structure is simple and the Kalman filter framework can be directly applied. Fang proposed an innovative adaptive extended Kalman filter method for the aerial alignment of airborne position and azimuth measurement systems [14]. This method improves the real-time performance of the algorithm and reduces the interference from GPS measurement noise. However, EKF can only achieve first-order accuracy and cannot handle strong nonlinear problems. In order to solve this problem, UKF is proposed. UKF assumes that the nonlinear system is a Gaussian probability density function (PDF) distribution and creates a series of sigma points based on the unscented transform rule to approximate the distribution of the system. It can achieve the second-order Taylor expansion polynomial. Saman Mukhtar Siddiqui applied the initialization process of the Central Difference Unscented Kalman filter (CDUKF) to the square-root unscented Kalman filter and verified the advantages of CDUKF in calculation time and accuracy in the on-board experiment to solve the alignment problem of misalignment angles within 30° [26]. Wang Dingjie proposed an adaptive unscented Kalman filter for a small UAV MEMS navigation system. With an initial error angle of 30°, the posture is completed in 75 s, which is more accurate than EKF and traditional UKF [27]. However, there is also a problem: For systems with a dimension higher than 3, the weight of the sigma points could be negative, and the filter process will be unstable. As a matter of fact, this problem is caused by the Unscented Transformation (UT) rule. To avoid the negativity of the sigma points, GHQF, SGQF, and CKF have been proposed using different integrated rules rather than the UT rule, which not only solves the mentioned problem but also maintains the accuracy of the second-order Taylor expansion. The GHQF utlizes the Gauss–Hermite quadrature (GHQ) rule to conduct the filter process with the quadrature sampling points that it can be accurate to the arbitrary order Taylor expansion polynomial [23]. However, the drawback of the GHQ rule is that it needs a large number of the sampling points due to the application of the direct tensor product. Thus, the GHQF would also have problems in application in high-dimensional systems. CKF with the cubature rule guarantees the positive weight, and its number of sampling points is 2n. It could also achieve a higher accuracy while applying the high-degree cubature rule with less computational burden [24]. Compared with CKF, the square-root CKF (SCKF) has more advantage not only in convergence speed but also in calculation efficiency [25]. To achieve a better performance of robustness during the in-flight process, we added several methods, such as Sage–Husa noise statistic and fading factors, to the frame of SCKF.

The rest of this paper is organized as follows. An Optimisation-Based initial Alignment (OBA) method based on the Rodrigues parameter for random misalignment angle is introduced in Section 2, where the state model and the measurement equation are used in the simulation. Then, the specific formulation of MSTASCKF is presented in Section 3. In Section 4, a semi-physical simulation is carried out to evaluate the performance of the proposed algorithm under the vibration condition. The alignment results are compared with the conventional SCKF. Finally, the conclusions are drawn in Section 5.

## 2. Alignment Model for Random Misalignment Angle

In this section, a dynamic alignment model in the form of a random misalignment angle [28] is introduced.

The coordinate systems are defined as follows:

Body frame (b-frame): The origin of the coordinate system is at the center of mass of the carrier, the *Y* axis is the longitudinal axis of the carrier, the *X* axis points to the right of the carrier and is perpendicular to the *Y* axis, and the *Z* axis and *X* and *Y* axes form a right-handed coordinate system.

Navigation frame (n-frame): The origin of the coordinate system is at the center of mass of the carrier, the *X* axis points to the geographic north direction, the *Y* axis points to the geographic east direction, and the *Z* axis points to the up direction.

Inertial navigation frame ($i_{n}$-frame): An inertial coordinate system, which coincides with the initial navigation frame.

Inertial body frame ($i_{b}$-frame): The inertial coordinate system, which coincides with the body frame at the initial time.

The attitude matrix can be decomposed into three parts based on the chain rule as follows:(1)$$C_{b}^{n}(t)=C_{in}^{n}(t)C_{ib}^{in}C_{b}^{ib}(t)$$
where $C_{in}^{n}(t)$
$C_{b}^{ib}(t)$ denotes the time-varying transformation matrix from the $i_{n}$-frame to the n-frame and the transformation matrix from the b-frame to the $i_{b}$-frame, respectively. $C_{ib}^{in}$ is a constant transformation matrix from the $i_{b}$-frame to the $i_{n}$-frame.

From Equation (1), we can see that the attitude matrix $C_{b}^{n}(t)$ can be divided into three parts:

$C_{in}^{n}(t)$ represents the time-varying attitude due to the rotation of the n-frame caused by the earth rotation rate and the vehicle’s transport rate related to earth, which can be calculated with the change of latitude and the longitude.

$C_{b}^{ib}(t)$ describes the time-varying attitude due to the rotation of the vehicle from the b-frame to the $i_{b}$-frame, which can be computed with the angular rate measured by the gyro. It is easy to derive the recursive form:(2)$$C_{b}^{ib}(t_{k})=C_{b}^{ib}(t_{k-1})C_{b(t_{k})}^{b(t_{k-1})},\;C_{b}^{ib}(0)=I$$

The $C_{b(t_{k})}^{b(t_{k-1})}$ represents the rotation of the b-frame from time $t_{k-1}$ to time $t_{k}$. It is appropriate to assume the rotation is tiny while the sampling rate of INS is high.
(3)$$C_{b(t_{k})}^{b(t_{k-1})}=[I-(-\omega_{b}(t_{k}))\times ]$$

As for the last part, $C_{ib}^{in}$ can be derived by solving the attitude determination problem using the relationship between vectors:(4)$$V^{in}=C_{ib}^{in}V^{ib}$$
where the $V^{in}$ and $V^{ib}$ are the integration of the specific force in the $i_{n}$-frame and the $i_{b}$-frame, respectively. We assume the specific force detected by accelerometer at time $t_{k}$ in the $i_{b}$-frame is given by following expression:(5)$${\hat{V}}^{ib}(t_{k})=V^{ib}(t_{k})+\delta V^{ib}(t_{k})$$
where the $\delta V^{ib}(t_{k})$ is the integration error of ${\hat{V}}^{ib}(t_{k})$.

Equation (4) could be rewritten as:(6)$$V^{in}(t_{k})=C_{ib}^{in}({\hat{V}}^{ib}(t_{k})-\delta V^{ib}(t_{k}))$$

The attitude matrix $C_{ib}^{in}$ can be written in the form of the Rodrigues vector [29] as:(7)$$C_{ib}^{in}={\left[I+\left(l\times \right)\right]}^{-1}\left[I-\left(l\times \right)\right]$$
where the l× is the skew-symmetric matrix of the Rodrigues vector.

Substituting Equation (7) to Equation (6) leads to:(8)$${\hat{V}}^{ib}(t_{k})-V^{in}(t_{k})=({\hat{V}}^{ib}(t_{k})+V^{in}(t_{k}))\times l+l\times \delta V^{ib}(t_{k})+w_{t_{k}}$$
where the $w_{t_{k}}$ is the inertial sensor error.

Defining the sum and difference as follows:(9)$$D_{t_{k}}={\hat{V}}^{ib}(t_{k})-V^{in}(t_{k}),\;S_{t_{k}}={\hat{V}}^{ib}(t_{k})+V^{in}(t_{k})$$

We could obtain:(10)$$D_{t_{k}}=S_{t_{k}}\times l+l\times \delta V^{ib}(t_{k})+w_{t_{k}}$$

Equation (10) is the measurement equation of our filtering model. Once the optimal estimation of the Rodrigues vector l is obtained, we can derive $C_{ib}^{in}$ from Equation (7).

Considering the bias and the random noise of the gyro and the accelerometer, the output of gyro and accelerometer can be derived as:(11)$${\hat{\omega }}^{b}=\omega^{b}+\varepsilon^{b}+w_{g}^{b}$$
(12)$${\hat{f}}^{b}=f^{b}+\nabla^{b}+w_{a}^{b}$$

Referring to the error model of INS under n-frame, the error equation of the attitude and velocity can be obtained:(13)$${\dot{\varphi }}^{ib}=-{\hat{C}}_{b}^{ib}(\varepsilon^{b}+w_{g}^{b}),\varphi^{ib}(0)=0$$
(14)$$\delta {\dot{V}}^{ib}=-{\hat{f}}^{ib}\times \varphi^{ib}+{\hat{C}}_{b}^{ib}(\nabla^{b}+w_{a}^{b}),\delta {\dot{V}}^{ib}(0)=0$$

The gradient of the Rodrigues vector is zero, since the attitude matrix $C_{ib}^{in}$ is a constant matrix.
(15)$$\dot{l}=0$$

The gyroscope drifts $\varepsilon^{b}$ and the accelerometer bias $\nabla^{b}$ are irrelevant with time, so we can derive:(16)$$\begin{array}{l} \varepsilon^{b}=0\\ \nabla^{b}=0 \end{array}$$

The state vector was selected as follows:(17)$$X={\left[\begin{array}{ccccc} l & \varphi^{ib} & \delta V^{ib} & \varepsilon^{b} & \nabla^{b} \end{array}\right]}^{T}$$

## 3. Mstasckf Method

In this section, we demonstrate the algorithm design ideas and the overall process of the algorithm. The algorithm was based on the frame of square-root Kalman filter and the Sage–Husa noise estimator. The aim to our work was to add a criterion to the Sage–Husa noise estimator to ensure the calculated system error variance remains positive, which could make the filter stable. Besides that, a multiple fading factor was implemented to make the residual orthogonal, which could accelerate system convergence and improve the accuracy.

### 3.1. Sage–Husa Noise Estimator

The Sage–Husa filtering method was originally proposed by P.A Sage and G.W Husa in 1969 [30]. The core idea of the algorithm is to construct a real-time filtered noise statistic based on maximum a posteriori information (MAP) to count the system noise and measurement noise in the filtering process.

For a nonlinear system as shown in (18):(18)$$\left\{\begin{array}{c} x_{k}=f\left(x_{k-1}\right)+w_{k-1}\\ z_{k}=h\left(x_{k}\right)+v_{k} \end{array}\right.$$
where the $w_{k-1}$ and $v_{k}$ are the system noise and measurement noise, respectively. $f\left(*\right)$ and $h\left(*\right)$ denote the nonlinear state model and measurement model, respectively.

Definite the system’s posterior error variance matrix at the initial moment as:(19)$$P_{0}=E\left[\left(x_{0}-{\bar{x}}_{0}\right){\left(x_{0}-{\bar{x}}_{0}\right)}^{T}\right]$$
where the ${\bar{x}}_{0}$ is the mean value of the initial state vector $x_{0}$. At time $t_{k}$, assuming the system noise $q_{k}$, the measurement noise $r_{k}$, the system noise covariance matrix $Q_{k}$, and the measurement noise covariance matrix $R_{k}$ are unknown. The state quantity set and quantity measurement set from the initial time to k time are, respectively, $X_{k}=\left[x_{0},x_{1},\cdots ,x_{k}\right]$, $Z_{k}=\left[z_{1},z_{2},\cdots ,z_{k}\right]$.

It can be seen from the Bayes formula that the probability of estimating the state vector at time $t_{k}$, the system noise, the measurement noise, and its noise variance matrix from the measurement sequence from the initial time to time $t_{k}$ can be expressed as:(20)$$\begin{array}{l} p\left(X_{k},q_{k},Q_{k},r_{k},R_{k}|Z_{k}\right)=J\\ =p\left(Z_{k}|X_{k},q_{k},Q_{k},r_{k},R_{k}\right)p\left(X_{k}|q_{k},Q_{k},r_{k},R_{k}\right)\times p\left(Z_{k}\right) \end{array}$$
where $p\left(Z_{k}\right)$ has nothing to do with noise estimation. Then, (20) can be reduced to the problem of solving the maximum value of probability density as follows:(21)$$p\left(Z_{k}|X_{k},q_{k},Q_{k},r_{k},R_{k}\right)\times p\left(X_{k}|q_{k},Q_{k},r_{k},R_{k}\right)\times p\left(q_{k},r_{k},Q_{k},R_{k}\right)$$

Considering that $Cov\left(q_{k},r_{k}\right)=0$ and each measurement can be regarded as independent.
(22)$${\hat{q}}_{k}=\frac{1}{k}\sum_{j=1}^{k}\left[{\hat{x}}_{j|k}-f_{j-1}\left(x_{j-1}\right)\right]$$
(23)$${\hat{Q}}_{k}=\frac{1}{k}\sum_{j=1}^{k}\left[{\hat{x}}_{jk}-f_{j-1}\left(x_{j-1}\right)-q_{k}\right]\times {\left[{\hat{x}}_{jk}-f_{j-1}\left(x_{j-1}\right)-q_{k}\right]}^{T}$$
(24)$${\hat{r}}_{k}=\frac{1}{k}\sum_{j=1}^{k}\left[z_{j}-h\left(x_{j}\right)\right]$$
(25)$${\hat{R}}_{k}=\frac{1}{k}\sum_{j=1}^{k}\left\{\left[z_{j}-h\left({\hat{x}}_{j|j-1}\right)-r_{k}\right]\times {\left[z_{j}-h\left({\hat{x}}_{j}\right)-r_{k}\right]}^{T}\right\}$$

When the noise changes with time, the filter estimates and measurements at the more recent time can better reflect the current noise situation. Therefore, it is necessary to adjust the weights of the filter estimates and measurements at different times in the noise statistics process to obtain more accurate noise statistics.

Define the weighting coefficient set $\left\{\lambda_{i}\right\}$, $\lambda_{i}=\lambda_{i-1}b$, $d_{k}=\frac{1-b}{1-b^{k}}$, i=0,1,2,…,k, b as a constant value between 0 to 1. Hence, we can derive a discrete noise estimator for time-varying noise as:(26)$${\hat{q}}_{k}=\left(1-d_{k}\right){\hat{q}}_{k-1}+d_{k}\left({\hat{x}}_{k}-\Phi_{klk-1}X_{k-1}\right)$$
(27)$${\hat{r}}_{k}=\left(1-d_{k}\right){\hat{r}}_{k-1}+d_{k}\left(z_{k}-H_{k}{\hat{X}}_{k/k-1}\right)$$
(28)$${\hat{R}}_{k}=\left(1-d_{k}\right){\hat{R}}_{k-1}+d_{k}\left(v_{k}v_{k}^{T}-H_{k}P_{k|k-1}H_{k}^{T}\right)$$
(29)$${\hat{Q}}_{k}=\left(1-d_{k}\right){\hat{Q}}_{k-1}+d_{k}\left(K_{k}v_{k}v_{k}^{T}K_{k}^{T}+P_{k}-\Phi_{k/k-1}P_{k-1}\Phi_{k/k-1}^{T}\right)$$
where $\Phi_{klk-1}$ is the one-step prediction matrix from time *k* − 1 to time *k* and H*_k_* is the measurement model of the system. $v_{k}=z_{k}-H_{k}{\hat{X}}_{k|k-1}-r_{k}$ is the residual of the measurement updating process.

### 3.2. SCKF with Sage–Husa Noise Estimator

There is a problem when applying the Sage–Husa noise estimator to the SCKF frame. Under the framework of square-root volume Kalman filtering, it is necessary to use the Cholesky decomposition form of the system noise variance matrix and the measurement noise variance matrix to perform filtering iterations during the filtering process [24], which requires us to always obtain positive definite in the noise estimator. Otherwise, the filter will be unstable.

Define the second part of (29) and (28) as:(30)$$J_{q}=K_{k}v_{k}v_{k}^{T}K_{k}^{T}+P_{k}-\Phi_{k/k-1}P_{k-1}\Phi_{k/k-1}^{T}$$
(31)$$J_{r}=v_{k}v_{k}^{T}-H_{k}P_{k|k-1}H_{k}^{T}$$

When the initial system noise covariance is positive, the sign of ${\hat{Q}}_{k}$ is only related to the sign of $J_{q}$, and the same situation occurs for ${\hat{R}}_{k}$. To ensure the stability of the filter, we used the equation as follows to compute ${\hat{Q}}_{k}$ and ${\hat{R}}_{k}$ in the condition $J_{q}<0$ or $J_{r}<0$.
(32)$${\hat{Q}}_{k}=\left(1-d_{k}\right){\hat{Q}}_{k-1}+d_{k}\left(K_{k}v_{k}v_{k}^{T}K_{k}^{T}+P_{k}\right)$$
(33)$${\hat{R}}_{k}=\left(1-d_{k}\right){\hat{R}}_{k-1}+d_{k}v_{k}v_{k}^{T}$$

Unfortunately, when the covariance matrix is computed with (32) or (33), the result is no longer unbiased. Thus, to make up to the accuracy loss caused by the bias, a multiple fading factor was introduced.

### 3.3. Multiple Fading Factor Calculation

The multiple fading factors method is based on the orthogonality principle of the residual. The computation process of multiple fading factors under SCKF could be summarized as follows, and readers can refer to the methodology described by the authors of [31] for more analysis and additional derivations.

Definite the multiple fading factor sequence as:(34)$$\gamma_{k}=diag[\gamma_{k}{}^{1},\gamma_{k}{}^{2},\ldots ,\gamma_{k}{}^{n}]$$
where *n* is the dimension of the state vector.
(35)$$\gamma_{k}^{i}=\left\{\begin{array}{ll} \gamma_{k}^{0i}, & \gamma_{k}^{0i}>1\\ 1, & \gamma_{k}^{0i}\leq 1 \end{array}\;\;\;i=1,\cdots ,n\right.$$
(36)$$\gamma_{k}^{0i}=\frac{\mathrm{tr}\left[N_{k}\right]}{\sum_{i=1}^{n}\alpha_{i}M_{k}^{in}}$$
(37)$$N_{k}=V_{k}-{\left[P_{xz,k|k-1}^{\left(l\right)}\right]}^{T}{\left[{\left(P_{k|k-1}^{\left(l\right)}\right)}^{-1}\right]}^{T}Q_{k-1}\left[{\left(P_{k|k-1}^{\left(l\right)}\right)}^{-1}\right]\left[P_{xzk|k-1}^{\left(l\right)}\right]-R_{k}$$
(38)$$M_{k}=\left(P_{k|k-1}^{\left(l\right)}-Q_{k-1}\right)\left[{\left(P_{k|k-1}^{\left(l\right)}\right)}^{-1}\right]\left[P_{xz,k|k-1}^{\left(l\right)}\right]{\left[P_{xz,k|k-1}^{\left(l\right)}\right]}^{T}{\left[{\left(P_{k|k-1}^{\left(l\right)}\right)}^{-1}\right]}^{T}$$

### 3.4. MSTASCKF Update

Step 1: Initialization
(39)$$S_{k-1}=chol\left(P_{k-1}\right)$$
where $S_{k-1}$ is the lower triangle matrix of the result of the Cholesky decomposition.

Step 2: Prediction

In the prediction step, MSTASCKF computes the mean $x_{k|k-1}$ and $S_{k|k-1}^{\left(l\right)}$, the square-root form of the associated covariance without the correction of the multiple fading factors, using the cubature rule, in which each cubature point has the same weight.

Obtain the cubature points:(40)$$X_{i,k-1|k-1}=S_{k-1}\xi_{i}+x_{k-1}$$

Propagate the cubature points through the state model:(41)$$X_{i,k|k-1}^{*}=f\left(X_{i,k-1|k-1}\right)$$

Estimate the predicted state:(42)$$x_{k|k-1}=\sum_{i=1}^{m}\omega_{i}X_{i,k|k-1}^{*}$$
(43)$$X_{k|k-1}^{*}={\left(\sqrt{\omega_{i}}\left(X_{i,k|k-1}-x_{k|k-1}\right)\right)}_{i=1}^{m}$$

In the proposed algorithm, the weights of the cubature points are equal, which is $\omega_{i}=\frac{1}{n}$.

Square-root factor of the predicted error covariance:(44)$$S_{k|k-1}^{\left(l\right)}=qr\left(\begin{array}{cc} X_{k|k-1}^{*} & S_{Q_{k-1}} \end{array}\right)$$
where $S_{Q_{k-1}}$ is the square-root of the system noise $Q_{k-1}$ at time *k* − 1 and $qr\left(\right)$ is the QR decomposition function to obtain the square-root factor of the matrix.

Step 3: Calculating the multiple fading factor

Obtain the cubature points:(45)$$X_{i,k|k-1}^{\left(l\right)}=S_{k|k-1}^{\left(l\right)}\xi_{i}+x_{k|k-1}$$

Propagate the cubature points through measurement model:(46)$$Z_{i,k|k-1}^{*(l)}=h\left(X_{i,k|k-1}^{\left(l\right)}\right)$$

Obtain the predicted measurement:(47)$$z_{k|k-1}^{\left(l\right)}=\sum_{i=1}^{m}\omega_{i}Z_{i,k|k-1}^{*\left(l\right)}$$
(48)$$Z_{k|k-1}^{*\left(l\right)}={\left(\sqrt{\omega_{i}}\left(Z_{i,k|k-1}^{\left(l\right)}-z_{k|k-1}^{\left(l\right)}\right)\right)}_{i=1}^{m}$$
(49)$$S_{zz,k|k-1}^{\left(l\right)}=qr\left(\begin{array}{cc} Z_{k|k-1}^{*\left(l\right)} & S_{R_{k}} \end{array}\right)$$
(50)$$P_{xz,k|k-1}=\chi_{k|k-1}^{\left(l\right)}Z_{k|k-1}^{*\left(l\right)}{}^{T}$$

Obtain the multiple fading factor with (35)–(38) and adjust $S_{k|k-1}$ with multiple fading factors:(51)$$S_{k|k-1}-\sqrt{\gamma_{k}}S_{k|k-1}^{\left(l\right)}$$

The l in the upper right corner of the symbol indicates that this quantity has not been corrected by multiple fading factors.

Step 4: Measurement updating

Create cubature points:(52)$$X_{i,k|k-1}=S_{k|k-1}\xi_{i}+x_{k|k-1}$$

Propagate the cubature points through measurement model:(53)$$Z_{i,k|k-1}^{*}=h\left(X_{i,k|k-1}\right)$$
(54)$$z_{k|k-1}=\sum_{i=1}^{m}\omega_{i}Z_{i,k|k-1}^{*}$$
(55)$$Z_{k|k-1}^{*}={\left(\sqrt{\omega_{i}}\left(Z_{i,k|k-1}-z_{k|k-1}\right)\right)}_{i=1}^{m}$$
(56)$$S_{zz,k|k-1}=qr\left(\begin{array}{cc} Z_{k|k-1}^{*} & S_{R_{k}} \end{array}\right)$$
(57)$$\chi_{k|k-1}={\left(\sqrt{\omega_{i}}\left(X_{i,k|k-1}-x_{k|k-1}\right)\right)}_{i=1}^{m}$$
(58)$$P_{xz,k|k-1}=\chi_{k|k-1}Z_{k|k-1}^{*}{}^{T}$$

Obtain the Kalman gain:(59)$$W_{k}=P_{xz,k|k-1}/S_{zz,k|k-1}/S_{zz,k|k-1}{}^{T}$$

Update the state estimation:(60)$$x_{k}=x_{k|k-1}+W_{k}\left(z_{k}-z_{k|k-1}\right)$$

Obtain the square-root of noise covariance estimation:(61)$$S_{k}=qr\left(\begin{array}{cc} \chi_{k|k-1}-W_{k}Z_{k|k-1}^{*} & W_{k}S_{R_{k}} \end{array}\right)$$

Step 5: Estimation of the noise covariance

Obtain $J_{q}$ and $J_{r}$.

Compute the system noise covariance and the measurement noise covariance based on the sign of $J_{q}$ and $J_{r}$ using (28), (29) or (32), and (33).

## 4. Semi-Physical Simulation Results and Discussion

In order to assess the performance of the MSTASCKF during the in-flight alignment for GPS/MIMU under vibration, a semi-physical simulation is described in this section. A trajectory maintaining the IMU data and GPS data of typical short-range guided weapon was created, and an experiment to simulate the flight vibration was carried out to obtain simulated vibration noise data. Then, the vibration noise was added to the IMU data to simulate the vibration in-flight. The performance of MSTASCKF was compared with the traditional SCKF, which validated the efficiency of the two strategies added in the MSTASCKF.

### 4.1. Vibration Experiment

Guided weapons generally experience storage, transportation, and flight environments after their production cycle. During flight, the high-speed turbulent flow field formed around the projectile due to high-speed flight induces complex vibration, overload, high temperatures, and other environments [32]. The vibration environment has a greater impact on the structure of the projectile and may even cause deformation of the projectile or shedding of solder joints on the circuit board. The vibration of guided weapons during flight can be divided into two categories: 0–2000 HZ, low-frequency noise excited by vibration mainly transmitted by mechanical structures, and 10–10,000 HZ noise excited by frequency range. Among them, vibration excitation noise has an obvious effect on the mass-spring structure, and excitation noise mainly has an obvious effect on the plate and shell structure. Therefore, the main focus of this paper was the impact characteristics of the frequency range 0–2000 HZ by vibration excitation noise on the output of the inertial navigation system.

In order to obtain the output data of MIMU under the vibration environment, the following random vibration test was designed. The power spectrum density and frequency range of the vibration are as shown in the Figure 1. The experimental platform is shown in Figure 2, which is composed of a MIMU (MTI-1 from the XSENS company), a vibration table, and the tooling for the vibration table. The specification of the MIMU is listed in Table 1.

Since the MTI-1 MIMU cannot be sensitive to the Earth’s rotation angular rate, its gyro output under static conditions can be regarded as zero bias and random noise. Vibration noise can be obtained by subtracting the average value of the data at rest when the vibration table is turned on. Thus, the process of the experiment was arranged as in Figure 3 to obtain both the static output and the vibration output of the three axes. The processed vibration noise data in the experiment are shown in Figure 4.

### 4.2. Simulation Condition

Generally, the flight time of short-range guided weapons is within 60 s, and the flight distance is within 20 km. According to the trajectory characteristics of short-range guided weapons, the trajectory attitude and speed used in the semi-physical simulation are shown in the Table 2. The attitude and the velocity of the trajectory are shown in Figure 5.

The trajectory was set for 30 s, and the initial attitude angle was set as $\psi =60^{\circ },\theta =45^{\circ },\gamma =0^{\circ }$, representing the yaw, pitch, and roll, respectively. The yaw change was positive from north to east and negative from west to north, the pitch change was positive from the horizontal section of the projectile upward and negative from the downward direction, the hour hand was positive and the counterclockwise was negative, and the projectile speed was the projectile speed at the end of the current time period.

In the simulation, the three-axis gyro data were all added with a 100°/h bias. The speed error of the GPS was 0.1 m/s. Thus, the initial state covariance was set as follows.
(62)$$\left\{\begin{array}{c} P_{0}=P_{0}⊗I_{3}\\ P_{0}=diag\left(\left[P_{l},P_{\varphi^{i_{b}}},P_{\delta V^{i_{b}}},P_{\varepsilon },P_{\nabla }\right]\right) \end{array}\right.$$
where $P_{l_{j}}={\left(1\right)}^{2}$, $P_{\varphi_{j}^{i_{b}}}={\left(1e-4^{\prime }\right)}^{2}$, $P_{\delta V_{j}^{j_{b}}}={\left(0.1\;m/s\right)}^{2}$, $P_{\varepsilon_{j}}={\left(100^{{}^{\circ}}/h\right)}^{2},P_{\nabla_{j}}={\left(3\;\mathrm{mg}\right)}^{2}$, $R=\mathrm{diag}\left({\left[\begin{array}{lll} 0.1 & 0.1 & 0.1 \end{array}\right]}^{2}\right)$.

### 4.3. Simulation Results under Vibration Noise

The in-flight alignment progresses were calculated using MSTASCKF and traditional SCKF, respectively. The attitude error is shown in Figure 6. The specific value of the estimated attitude error and its RMS of the proposed method and SCKF is shown in Table 3.

The figure shows that the MSTASCKF has an advantage in convergence speed in the alignment process. The estimation of yaw, pitch, and roll converged within 15 s and finally reached the accuracy of 0.08°, 0.05°, and 0.25°, respectively. The SCKF had a worse performance, which can be easily observed as it converged slower and only met the accuracy of −2.85°, 0.95°, 1.65° for the yaw, pitch, and roll under the vibration noise.

To further compare the stability of the MSTASCKF and SCKF, the RMS of the attitude is shown in the Figure 7.

We can see that the proposed method had a better RMS performance than the SCKF. The RMS values of the three attitudes of MSTASCKF and SCKF at the end of the alignment process were $\left[\begin{array}{ccc} 1.13^{\circ } & 0.14^{\circ } & 5.35^{\circ } \end{array}\right]$ and $\left[\begin{array}{ccc} 2.5^{\circ } & 0.99^{\circ } & 6.12^{\circ } \end{array}\right]$, respectively. The RMS of the yaw and roll was much larger than the pitch. This was caused by the estimation scheme, since the MSTASCKF and SCKF had the same trend. The first estimation of roll and pitch had such a large deviation that the subsequent RMS value was affected.

In fact, the main metrics of alignment performance was the final accuracy at the end of the alignment process rather than the accuracy during the process.

The attitude error is shown in the Table 4 when the alignment time was limited to 10 s and 20 s. These data could also show that the MSTASCKF had an advantage in the convergence speed and the estimation stability.

### 4.4. Discussion

In this section, a semi-physical simulation for in-flight alignment under vibration noise was carried out to evaluate the performance of the proposed algorithm. Compared with conventional SCKF, it is apparent that the proposed method had a better performance not only in the convergence speed in the beginning of the process but also in the accuracy and steady state. For the proposed filter, the results are as follows.

We proposed a robust SCKF algorithm, which could significantly increase the in-flight alignment accuracy under the vibration noise.

Compared with conventional SCKF filter, the estimation performance was better. The attitude error was smaller and the RMS was also smaller. The convergence time of the proposed method was no longer than 10 s. Thus, the proposed method can meet the rapidity and accuracy requirements of in-flight alignment.

## 5. Conclusions

The rapid and accurate in-flight alignment has a vital impact on the performance of the guided weapon applied MIMU. To improve the in-flight alignment performance of MIMU/GPS system under flight vibration, this paper proposed a MSTASCKF filter based on the Sage–Husa noise estimator and multiple fading factors. The filter solved the instability problem of SCKF caused by the Sage–Husa estimator and improved the accuracy. The main work of this paper was as follows.

An experimental vibration environmental test was conducted to maintain the simulated vibration noise data during the flight and a semi-physical simulation was carried out to evaluate the performance of the proposed algorithm. The filtering method had an advantage in the convergence speed, accuracy, and stability compared to SCKF.

## Figures and Tables

**Figure 1 sensors-21-02597-f001:**
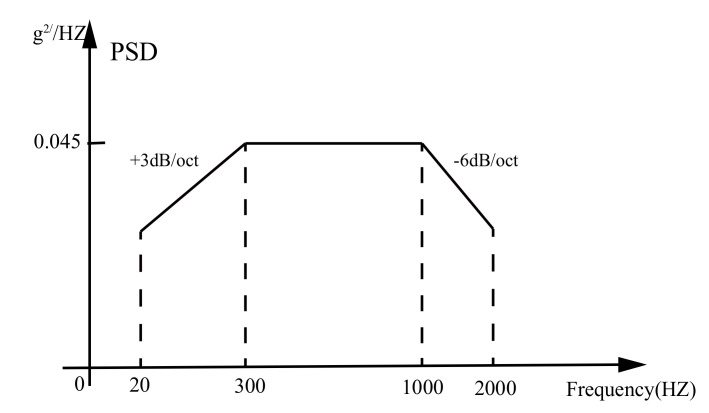
The power spectrum density and frequency range of the vibration experiment.

**Figure 2 sensors-21-02597-f002:**
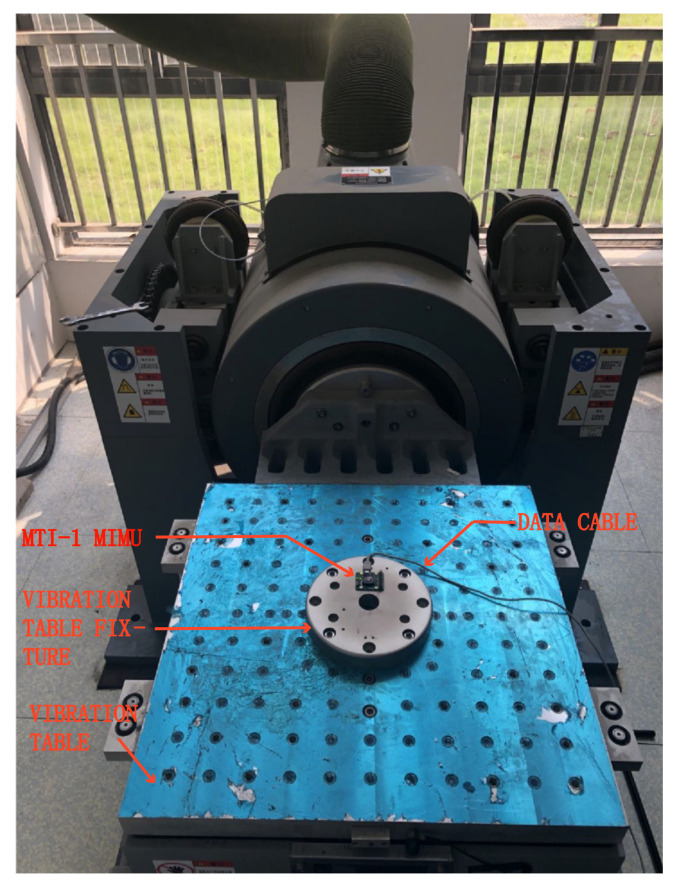
The experimental platform.

**Figure 3 sensors-21-02597-f003:**
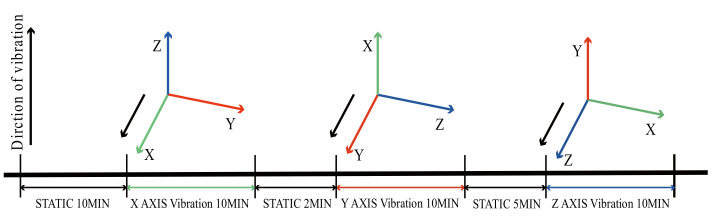
The vibration experiment process.

**Figure 4 sensors-21-02597-f004:**
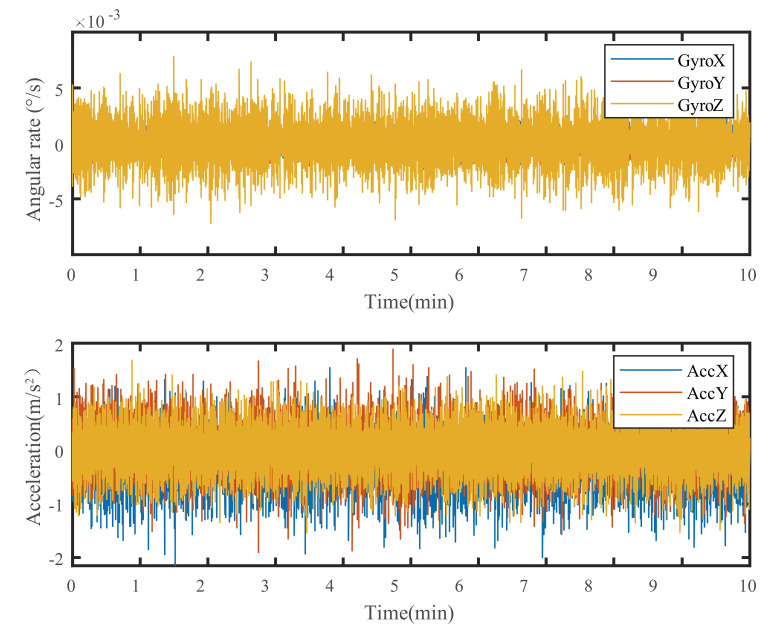
The vibration noise data of the vibration experiment of each axis.

**Figure 5 sensors-21-02597-f005:**
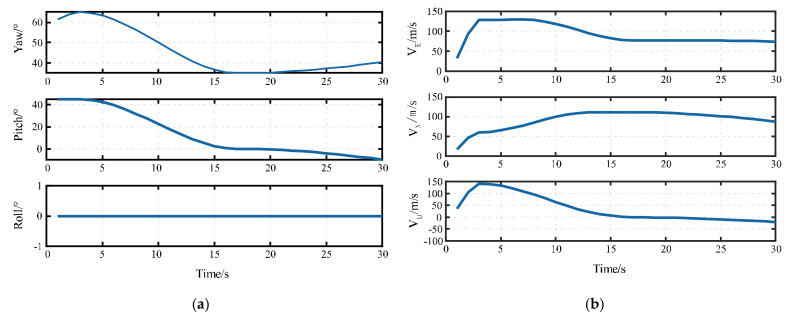
The real-time trajectory of the simulation. (**a**) The real-time attitude of the simulated trajectory. (**b**) The real-time velocity of the simulated trajectory in the direction of east, north, and up, respectively.

**Figure 6 sensors-21-02597-f006:**
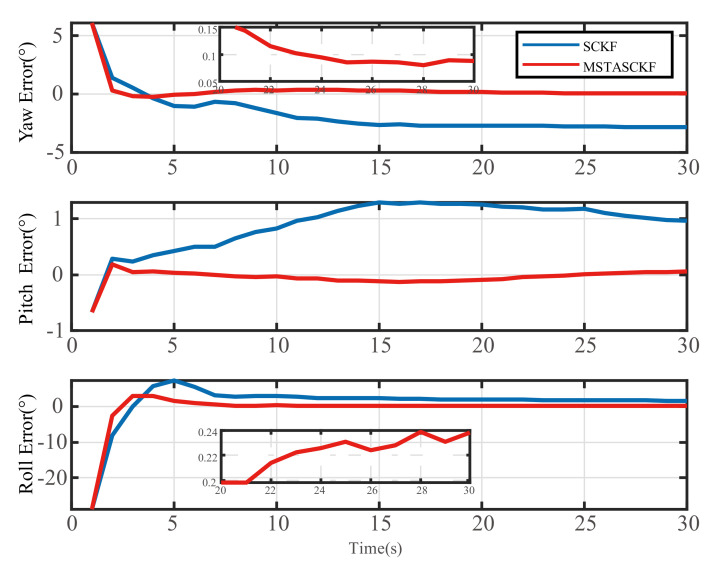
The attitude error comparison between MSTASCKF and SCKF.

**Figure 7 sensors-21-02597-f007:**
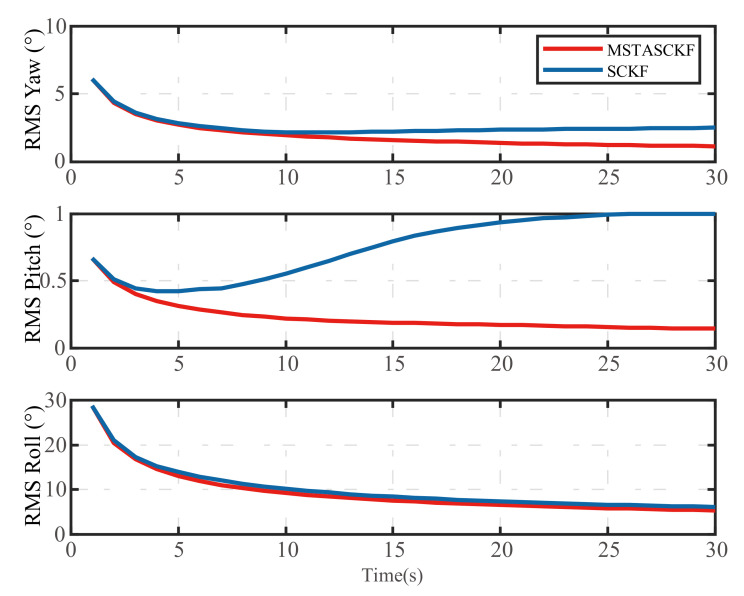
The RMS of the attitude error of MSTASCKF and SCKF.

**Table 1 sensors-21-02597-t001:** Specifications of MTI-1.

Parameters	Accelerometer	Gyroscope
Dynamic range	±16 g	±2000°/s
Update rate	100 Hz	100 Hz
Bias	3 mg	0.15°/s
Bias stability	0.03 mg	10°/h
Random walk	120 μg/√Hz	0.07°/s/√Hz

**Table 2 sensors-21-02597-t002:** The specific value of the attitude and velocity used in the trajectory.

Time	Yaw Change	Pitch Change	Roll Change	Speed
1–3 s	5°	0°	0	200 m/s
4–17 s	−30°	−45°	0	135 m/s
18–30 s	25°	−45°	0	50 m/s

**Table 3 sensors-21-02597-t003:** The attitude error and RMSE comparison between MSTASCKF and SCKF.

	MSTASCKF		SCKF	
	Attitude Error	RMS	Attitude Error	RMS
Yaw	0.08°	1.13°	−2.85°	2.5°
Pitch	0.05°	0.14°	0.96°	0.99°
Roll	0.25°	5.35°	1.66°	6.12°

**Table 4 sensors-21-02597-t004:** Thea attitude error comparison between MSTASCKF and SCKF when the alignment time was 10 s and 20 s.

	Attitude Error			
	MSTASCKF		SCKF	
Time	10 s	20 s	10 s	20 s
Yaw	0.29°	0.15°	−1.60°	−2.73°
Pitch	−0.02°	−0.08°	0.82°	1.25°
Roll	0.32°	0.19°	2.89°	2.05°

## Data Availability

Not applicable.

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
