# Peer review of "Robust SCKF Filtering Method for MINS/GPS In-Motion Alignment"

_sensors, 2021, doi:10.3390/s21082597_

Round 1

Reviewer 1 Report

This paper presents a novel multiple strong tracking adaptive sqaure-root cubature filter (MSTASCKF) based on the frame of Sage-husa filter employing multi-fading factor. The proposed algorithm is tested and confirmed via simulation and experiments. The overall approach is interesting. However, there are some questions and suggestions should be accounted.

1、In the introduction, a lot of literatures about different filtering algorithms are listed, but the research results about in-flight alignment are lacking. It is suggested to supplement the relevant research contents.

2、The state vectors listed in equation (16) include gyroscope drift and accelerometer bias. It is suggested to list the corresponding state equations to improve the integrity of the paper.

3、The variables or parameters in many formulas are not explained clearly in this paper.

For example, the meanings of f(xk-1)and h(xk-1)in formula (17) are not explained. In formula (28), the meaning of dk and its value are not explained. The meaning of a and its value in formula 39, are not explained. Similarly, is the weight coefficient in formula (41) equal weight? In order to improve the clarity of the paper, it is suggested that the author check the full text and explain the relevant variables and parameters.

4、The trajectory design in Table 2 should correspond to figure 5 (a), but the description in the table and the figure do not seem to correspond. The time settings 14s and 13s in the table do not correspond to the figure. The setting of pitch angle at 3S in the table does not correspond to that in the figure. The speed setting in the table does not indicate which direction of the speed setting in the figure. Please amend or explain.

5、In Table 1, the gyroscope bias is set to 0.15°/s, however, in the simulation, Pɛj=(100°/h)2, please explain the reason for this setting.

6、It is suggested to supplement vehicle experiments to verify the effectiveness of the algorithm under different maneuvering conditions.

7、There are some technical or grammatical unclearness or errors that need to be clarified or corrected. Please check the whole paper more carefully.i.e.,”GHQF would also has…”in the introduction.

8、Please check the labels of the figures and the labels of the tables. For example, figure 6 is marked twice.

Reviewer 2 Report

The manuscript presents a filtering method for MEMS-INS/GPS attitude estimation using an adaptative Square-Root Cubature Kalman Filter in order to adjust the process noise a priori estimation in a high vibration scenario.

The authors correctly introduce the topic and context of the study. They clearly detail the problematic of attitude estimation in the context of guided weapon due to high vibration patterns. They also mention the limitations in using a MEMS IMU with GPS in a KF solution.
Finally, multiple references are listed and highlighted in a short state of the art review on the existing mathematical solutions and their limitations.

The second part of the manuscript is a description of the alignment model under random misalignment angle. The mathematical model is herein presented and described.

The third part of the manuscript is a description of the designed algorithm process. The Square-Root Kalman Filter and the sage-husa noise estimator are presented.
However, some new elements could be added to this part of the manuscript by the authors to provide a better understanding of the mathematical model. Even if correct references are listed and introduced, a reminder of the details of the equations should be provided in the current manuscript.

Here are some more detailed comments on part 3:
A minor improvement could concern the description of expressions of the noise estimators given by formulas 25, 26, 27 and 28 in Lines 201 to 204 in part 3.1. Some of the terms in these equations are not explained and cannot be correctly understood for a reader. I advise the authors to provide some explanations on these equations by adding a detail of the variables.
In part 3.2, I have similar comments regarding the lack of explanations or details about some terms and variables described in the current part. This follows the lack of details given on the equations of the sage-husa noise estimator. For instance, no introduction of Sq and d.

The fourth part of the manuscript presents the simulations results and described in detail the proposed scenario. The experimental platform and the process to generate vibration patterns is also described.
The description of the scenario and simulation is correctly provided by giving some tables and a figure of the simulated trajectory.
Finally, the attitude estimation results are provided with a comparison between the proposed solution and the standard SCKF.
The comparison of the attitude estimation error at the end of the simulation is choose to compare with the SCKF solution and concludes on the performance improvement of the proposed solution.
In a second time, the RMS criteria is proposed to compare the two solutions. However, some details on the computation of the RMS should be provided like the number of simulations completed for this RMS values for instance.

Some typos, misspelling or missing information are however present in the manuscript and some of them are listed:

- Lines 98 to 101: Is the name of the in and ib frames is really chosen to be inertial navigation frame or initial navigation frame ?
- Lines 105 to 108: I think there is an error in this part because the detail of the terms of equation (1) is not clear at all. It seems that a reading of this part by the authors should be done again.
- Line 167: Residual text from the manuscript model I think (“All figures and tables should be cited in the main text as Figure 1, Table 1, etc.”)
- Lines 235: some residual text seems to be here ? (There’s a problem to apply sage-husa method to SCKF frame. Define) This line doesn’t make sense for me.
- Lines 292 to 295. This phrase is not clear and needs reformulation.

Round 2

Reviewer 1 Report

No comment